# Immobilization of Dextranase Obtained from the Marine *Cellulosimicrobium* sp. Y1 on Nanoparticles: Nano-TiO_2_ Improving Hydrolysate Properties and Enhancing Reuse

**DOI:** 10.3390/nano13061065

**Published:** 2023-03-15

**Authors:** Yingying Xu, Huanyu Wang, Qianru Lin, Qingzhen Miao, Mingwang Liu, Hao Ni, Lei Zhang, Mingsheng Lyu, Shujun Wang

**Affiliations:** 1Jiangsu Key Laboratory of Marine Bioresources and Environment/Jiangsu Key Laboratory of Marine, Biotechnology, Jiangsu Ocean University, Lianyungang 222005, China; 2Co-Innovation Center of Jiangsu Marine Bio-Industry Technology, Jiangsu Ocean University, Lianyungang 222005, China

**Keywords:** dextranase, TiO_2_ nanoparticles, immobilization, hydrolysates

## Abstract

Dextranase is widely used in sugar production, drug synthesis, material preparation, and biotechnology, among other fields. The immobilization of dextranase using nanomaterials in order to make it reusable, is a hot research topic. In this study, the immobilization of purified dextranase was performed using different nanomaterials. The best results were obtained when dextranase was immobilized on titanium dioxide (TiO_2_), and a particle size of 30 nm was achieved. The optimum immobilization conditions were pH 7.0, temperature 25 °C, time 1 h, and immobilization agent TiO_2_. The immobilized materials were characterized using Fourier-transform infrared spectroscopy, X-ray diffractometry, and field emission gun scanning electron microscopy. The optimum temperature and pH of the immobilized dextranase were 30 °C and 7.5, respectively. The activity of the immobilized dextranase was >50% even after 7 times of reuse, and 58% of the enzyme was active even after 7 days of storage at 25 °C, indicating the reproducibility of the immobilized enzyme. The adsorption of dextranase by TiO_2_ nanoparticles exhibited secondary reaction kinetics. Compared with free dextranase, the hydrolysates of the immobilized dextranase were significantly different, and consisted mainly of isomaltotriose and isomaltotetraose. The highly polymerized isomaltotetraose levels could reach >78.69% of the product after 30 min of enzymatic digestion.

## 1. Introduction

Dextranase (EC.3.2.1.11) specifically hydrolyzes straight-chain α-1,6-glycosidic bonds [1], and the enzymatic products are polysaccharides and isomalto-oligosaccharide [2]. Dextranase is widely used in the sugar, food, and pharmaceutical industries, as well as in green manufacturing and biotechnology [3,4,5,6,7]. Immobilization of enzymes improves their applicability [8] and storage [9]. Several carriers, such as alginate [10], chitin [11], agarose [1], and hydroxyapatite [12], have been used to immobilize dextranase. Although the immobilization rate of the immobilized dextranase has been improved, the stability and reusability of the immobilized enzymes are still not adequate for practical applications; therefore, identifying suitable immobilized materials continues to be the focus of research [13,14].

Nanomaterials have been widely used in various scientific studies. Inorganic nanomaterials show some advantages in clear composition and easy preparation [15,16]. Nano-Ag and nano-Au might positively affect enzymes distribution after immobilization, reducing undesirable side effects and protecting enzymes from degradation [17,18]. Nano-Ag has a positive effect on skin healing and protection from UV radiation, so Agnieszka M. Pudlarz et al. investigated immobilization of antioxidant enzymes on gold and silver nanoparticles [19]. Platinum (Pt) nano was used to immobilize glucose oxidase for biosensor fabrication, and Pt-Ag, a biomaterial with an antimicrobial surface that inhibits or prevents bacterial growth, is a suitable material for enzyme immobilization [20,21]. Among them, metal oxide nanoparticles are of interest, owing to their ability to reduce mass transfer limitations [22], the possibility of being made in several shapes and sizes [23], and their stability and nontoxicity [24]. Their efficiency in immobilizing enzymes is substantially higher, owing to the size of the nanomaterials [25]. Al_2_O_3_ was used in immobilization applications such as laccase as well as xylanase, due to its highly specific surface area, porosity, chemical-mechanical stability [26], and mechanical resistance at high pH and temperature [27]. Nano SiO_2_ has excellent carrier capacity and adjustable morphological characteristics, while also being low-cost and easy to synthesize. Meanwhile, it contains a large number of hydroxyl groups on the surface, which facilitates functional modification [28]. Therefore, SiO_2_ is also a promising carrier, by increasing protein binding and improving the performance of immobilized enzymes [29,30]. Lixi Cai et al. used SiO_2_ to coat magnetic nanocarriers and immobilization of β-1,3-xylanase [31]. ZrO_2_ has excellent mechanical and chemical properties [32,33], and was a material used for immobilization. R. Reshmi et al. conducted a study regarding immobilization of amylase on ZrO_2_ [34]. Titanium dioxide (TiO_2_) has received the most attention because of its biocompatibility [35] and its ability to maintain its natural properties and conserve the biological activity of enzymes [36,37]. Furthermore, TiO_2_ is inexpensive, has antibacterial effects, and is mechanically strong and corrosion-resistant [38,39,40]. Therefore, in recent scientific studies, nano-TiO_2_ has often been considered a suitable carrier for enzyme immobilization [41,42,43]. However, there are only a few studies on the immobilization of dextranase using nano-TiO_2_.

In this study, we investigated the immobilization efficiency of dextranase on nano-TiO_2_ under different conditions. We also studied the changes in the enzymatic properties and hydrolysates of immobilized materials, and explored the reproducibility and storability of the immobilized dextranase.

## 2. Materials and Methods

### 2.1. Materials

Dextranase was purified from the fermented broth of *Cellulosimicrobium* sp. Y1, a dextranase-producing strain obtained by screening in our laboratory. Nano-Al_2_O_3_, nano-SiO_2_, nano-ZrO_2,_ and nano-TiO_2_ were purchased from Shanghai Suita Materials Technology Co., Shanghai, China. Dextran T70 was purchased from Sinopharm Chemical Reagent Co., Shanghai, China. Nano-Ag, nano-Au, nano-Pt-Ag, and nano-Pt were provided by Prof. Zhao, School of Chemical Engineering, Jiangsu Ocean University. Tris-HCl was purchased from BioFROXX, Einhausen, Germany. All other reagents were purchased from Sinopharm (Shanghai, China). and were of analytical purity. Ultrapure water was used for all experiments.

### 2.2. Methods

#### 2.2.1. Selection of Nanomaterials

Different inorganic nanomaterials (Al_2_O_3_, SiO_2_, Ag, TiO_2_, Au, ZrO_2_, Pt-Ag, and Pt) were selected for the immobilization of dextranase. The nanomaterials were mixed with dextranase in 2-mL Eppendorf tubes for immobilization, and the supernatant and precipitate were separated by centrifugation at 6200× *g* for 5 min. The resulting precipitate was washed with water three times. After washing, the precipitate was mixed with 500 μL of water, and the enzyme activity of the precipitate was determined using the 3,5-dinitrosalicylic acid (DNS) method. In total, 50 μL of enzyme solution and 150 μL of substrate (3% dextran T70) were placed in a water bath at 40 °C for 15 min. Then, 200 µL of 3,5-dinitrosalicylic acid preparation reagent (DNS) was added and then boiled in boiling water for 5 min. Three milliliters of distilled water were added to the resulting mixture, and in the control group, the enzyme solution was added after DNS. Each sample took 200 µL to detect at 540 nm (Multiskan Go, Shanghai Bajou Industrial Co., Shanghai, China). The data were used to calculate enzyme activity [44].

#### 2.2.2. Optimization of the Immobilization Process

##### Effect of TiO_2_ Content

TiO_2_ (15–45 mg) was weighed and washed three times with 1 mL of sterile water. Then, 2 mL of enzyme solution was added and fixed at 25 °C for 3 h. After immobilization, the solution was centrifuged at 6200× *g* for 5 min. The precipitate was washed with 1 mL of water, 3 times. The precipitate was dissolved in 500 µL of water and enzyme activity was determined.

##### Effect of Adsorption Time

Thirty-five micrograms of TiO_2_ nanoparticles were weighed and fixed with enzyme solution at 25 °C for 1–5 h. Then, the solution was centrifuged at 6200× *g* for 5 min. The precipitates were washed three times, and the enzyme activity of the precipitates was determined at different fixation times.

##### Effect of Temperature and pH

The pH of the enzyme solution was adjusted to 6, 7, and 8, using a 50 mM Tris-HCl buffer. The pH-adjusted enzyme solution was immobilized with nano-TiO_2_ at 25 °C for 3 h. Then, the solution was centrifuged at 6200× *g* for 5 min and the enzyme activity of the precipitate was measured. The enzyme solution (2 mL) was fixed with TiO_2_ for 1 h at 4 °C, 25 °C, 30 °C, and 45 °C, respectively. Next, the solution was centrifuged at 6200× *g* for 5 min, followed by the determination of enzyme activity.

#### 2.2.3. Properties of Immobilized Dextranase

##### Effect of Temperature on Enzyme Activity and Stability

The enzymatic activity of immobilized dextranase was determined at different temperatures. The enzyme solution was incubated at 35 °C, 40 °C, and 45 °C, and the enzyme activity after incubation was measured after holding for 1–3 h. The relative enzyme activities were also determined and compared at different temperatures.

##### Effect of pH on Enzyme Activity and Stability

Different pH conditions (sodium acetate buffer, pH 4.0~6.0; phosphate-buffered saline, pH 6.0~7.5; and Tris-HCl buffer, pH 7.5~9.0) were set. Dextranase was mixed with buffers having different pH, and incubated in a water bath at 25 °C for 1 h. Enzyme activity was determined, and the stability of enzymes under different pH conditions was confirmed.

#### 2.2.4. Characterization of Immobilized Dextranase

Nano-TiO_2_ and immobilized dextranase were ground into a powder using an agate mortar. The powder was compressed into thin discs for Fourier-transform infrared spectroscopy (FT-IR, Thermo Fisher Scientific, Waltham, MA, USA) in transmission mode in the mid-infrared range of 400–4000 cm^−1^. Crystallinity analysis of TiO_2_ nanoparticles was performed using X-ray diffraction (XRD, PANalytical B.V., Almelo, The Netherlands) in the diffraction range of 10° to 90° (2θ). The immobilized enzymes were washed with water three times, resuspended in 1 mL of water, and lyophilized at −80 °C. The nano-TiO_2_ was then analyzed using field emission gun scanning electron microscopy (FEG-SEM, FEI, Hillsboro, OR, USA), and the particle morphology of the nano-TiO_2_ immobilized dextranase was determined.

#### 2.2.5. Analysis of the Hydrolysates of Immobilized Dextranase

Oligosaccharide preparation: 5 mg of each of glucose, isomaltose, isomaltotriose, isomaltotetraose, isomaltopentaose, isomaltohexose, and isomaltoheptaose were weighed and dissolved in 1 mL of water. Then, they were filtered by a 0.45 μm microporous membrane, and sonicated for 30 min using an ultrasonic cleaner (Ningbo Xinzhi Freeze-Drying Equipment Co., Ningbo, China) at 300 W power. Sample preparation: 30 g/L dextran T70 substrate was mixed with the enzyme solution at 3:1 (v/v). Free dextranase was held at 40 °C; the immobilized dextrose enzyme was held at 30 °C for 30 min, 1 h, 2 h, 3 h, and 4 h each, then boiled for 5 min and filtered through a microporous membrane, before ultrasonication. The samples and standards were analyzed by HPLC (Agilent, Santa Clara, CA, USA), using the Waters 600 with a waters sugar-pak1 column (6.5 × 300 mm), a mobile phase of deionized water, a flow rate of 0.4 mL/min, a column temperature of 75 °C, and an injection volume of 20 µL. The composition and content of each standard and sample were analyzed according to the reported peak areas.

#### 2.2.6. Reproducibility and Storage of Immobilized Dextranase

To the immobilized enzyme, 400 µL of the optimum pH substrate was added, mixed thoroughly, and reacted for 15 min at 30 °C. The contents were centrifuged at 6200× *g* for 5 min. DNS was added to 200 µL of the supernatant, and the reaction was terminated by boiling. The enzyme activity was then measured. For repeated use, the substrate was continuously added to the precipitate to 400 µL, the above steps were repeated, and the enzyme activity was determined. The samples were placed at 25 °C, which was chosen as the storage temperature, and samples were drawn at regular intervals to determine enzyme activity.

#### 2.2.7. Adsorption Kinetics

The kinetic model was used to describe the pattern of changes that occurred between the adsorbent and the adsorbate during adsorption. The pseudo first-order (PFO) kinetic model proposed by Lagergren was used, which assumes that the adsorption rate is proportional to the number of unoccupied adsorption sites [45]. In contrast, the pseudo second-order (PSO) kinetic model proposed by Blanchard allows determination of whether the surface reaction is controlled by the rate [46].
(1)$$\mathrm{PFO}\;\mathrm{kinetic}\;\mathrm{model}\;ln\left(q_{e}-q_{t}\right)=lnq_{e}-K_{1}t$$
(2)$$\mathrm{PSO}\;\mathrm{kinetic}\;\mathrm{model}\;\frac{t}{q_{t}}=\frac{1}{K_{2}q_{e}{}^{2}}+\frac{1}{q_{e}}t$$
where *q_t_* (mg/g) is the adsorption amount at time *t*; *q_e_* (mg/g) is the equilibrium adsorption amount; *K*_1_ (1/min) is the primary adsorption rate constant; and *K*_2_ (g/mg·min) is the secondary rate constant.

#### 2.2.8. Data Analysis

All experiments were parallelly performed in triplicate, and the data were analyzed to determine significance.

## 3. Results and Discussion

### 3.1. Nanomaterial Selection

As shown in Figure 1, among the different nanomaterials that were considered, TiO_2_ nanoparticles showed the best immobilization effect. Among the different particle sizes of TiO_2_ (5 nm, 30 nm, 50 nm), 30 nm was found to be ideal.

The immobilization effect of different particle sizes on dextranase, varies greatly. TiO_2_ with a particle size of 50 nm exhibits very poor dextranase-immobilization effects. As nanomaterials, the larger the particle size, the smaller the surface area of the total materials [47,48], as immobilization was performed using the same weight of nano-TiO_2_. The lower immobilization effect of 5-nm TiO_2_ versus 30-nm TiO_2_ may be due to the spatial resistance that affects the immobilization of dextranase, resulting in less-effective immobilization when using the former.

### 3.2. Immobilization Conditions

Nano-TiO_2_ content affected immobilization efficiency, and the best efficiency was achieved when the nano-TiO_2_ content was 35 mg (Figure 2a). The optimum immobilization temperature was 25 °C (Figure 2b) and the reaction time was at least 1 h (Figure 2c). The optimal pH for immobilization was 7 (Figure 2d).

The increased amount of enzyme could improve the efficiency of enzyme fixation; however, it seemed that excessive enzyme might lead to aggregation, and reduce the efficiency of enzyme fixation products. The result was similar to the report of Wanich Suksatan et al. [49]. The optimum immobilization temperature of enzyme was 25 °C, which is commonly described as room temperature. The optimal immobilization time was 5 h; however, it stabilized after the immobilization time reached 1 h. Thus, 80% of dextranase can be immobilized after incubation with TiO_2_ nanoparticles for 1 h, greatly reducing the process time for future industrial applications.

### 3.3. Enzymatic Properties of Immobilized Dextranase

#### 3.3.1. The Effect of Temperature on Enzyme Activity and Stability

The optimum temperature for the activity of free dextranase is 40 °C. The optimum reaction temperature of dextranase changed to 30 °C after immobilization (Figure 3a). Figure 3b shows that the remaining enzyme activity can be maintained at above 80%, even after holding the enzyme at 30 °C and 40 °C for 2.5 h. Moreover, the remaining enzyme activity of the immobilized dextranase could still be maintained at over 70%, after holding it at 30 °C for 3 h. The thermostability of the immobilized dextranase decreased markedly.

The optimal active temperature of immobilized dextranase decreased to 30 °C from 40 °C, and it might be the result of changed charge of the surface of enzyme. Charge interaction could affect the stability of enzymes in mesoporous materials [50]. Enzyme adsorption could lead the reduction in polar amino acids on the enzyme surface exposed to the solution. Meanwhile, the stability of the immobilized enzyme was relatively decreased, due to the electrostatic repulsion between the enzyme and the carrier when the enzyme and the nanomaterials carry the same charge [51].

#### 3.3.2. Effect of pH on Enzyme Activity and Stability

The optimum pH of both free and immobilized dextranase was 7.5 (Figure 4). The immobilized dextranase exhibited good stability from pH 5 to 7.5. Immobilization significantly improved the stability under acidic conditions. Additionally, almost 50% of the enzyme activity was maintained under alkaline conditions.

The optimum pH was maintained at 7.5, however, the stability of immobilized dextranase had changed remarkably at pH 5–6, compared with the free enzyme. Our results are similar to the study of β-galactosidase immobilization with chitosan particles, in which the stability of the enzyme was greatly improved after immobilization [52]. Azra Shafi studied the immobilization of β-galactosidase in SiO_2_ and found a considerable expansion in the stability of the immobilized enzyme, which exhibited higher activity on both sides of the pH-optima [24].

### 3.4. Characterization of Immobilized Dextranase

Immobilized dextranase was characterized using FT-IR, XRD, and FEG-SEM. The results showed that dextranase could be immobilized on the surface of TiO_2_ nanoparticles. The spectra of the immobilized materials changed at approximately 1526.86 cm^−1^, 1377.83 cm^−1^, and 1299.89 cm^−1^ (Figure 5a). The bending vibration of N–H at 1526.86 cm^−1^ corresponds to the bending vibration of N–H, which causes a change in peak shape. With the adsorption of the enzyme at 1377.83 cm^−1^ and at 1299.89 cm^−1^, a new band appeared that belongs to the C–O and C–N stretching vibrations. This result indicated that a new carboxylic acid group was generated by enzyme immobilization [12,53]. Omnic 9.2 software was used to plot FT–IR spectrum from 800 cm^−1^ to 2000 cm^−1^, and the vibration band in the spectrum can be seen more clearly. See Appendix A for details. The crystal morphology of TiO_2_ nanoparticles before and after immobilization did not change, as seen in the XRD plots (Figure 5b). It indicated that the immobilization of dextranase did not affect the crystal morphology of TiO_2_ nanoparticles. Our results were similar to the immobilization of laccase on titanium dioxide nanostructures studied by Jantiya Isanapong et al. [23]. The immobilized materials showed a decrease in the pores between TiO_2_ nanoparticles and an increase in agglomeration (Figure 5c,d), indicating the successful immobilization of dextranase on TiO_2_ nanoparticles. Wanich Suksatan et al. had similar results in the study of dextro-anhydrase immobilization in novel Ni/Zn-based nanostructures [49].

### 3.5. Analysis of the Hydrolysates

Figure 6a shows the sequential peaks of the standard oligosaccharides isomaltoheptaose, isomaltohexose, isomaltopentaose, isomaltotetraose, isomaltotriose, isomaltose, and glucose. Compared with those of standard sugar, the hydrolysates of the free enzyme were mainly isomaltopentaose and isomaltotriose, whereas the hydrolysates of immobilized dextranase were isomaltotetraose and isomaltotriose (Figure 6b). The percentage of hydrolysates changed with the hydrolysis time; 78.69% of isomaltotetraose was detected in the hydrolysates after incubation for 30 min (Table 1). Figure 6c shows the hydrolysates obtained by immobilizing dextranase with different sizes of nano-TiO_2_. After incubation for 1 h, isomaltotriose and isomaltotetraose were the main products when 5 nm and 30 nm of nano-TiO_2_ were used, respectively. On the other hand, the hydrolysates of 50-nm TiO_2_-immobilized dextranase were similar to those of free dextranase, and consisted of isomaltotriose, isomaltotetraose, and isomaltopentaose. Our results showed that the predominant hydrolysate of immobilized dextranase was isomaltotetraose, which was significantly different from that obtained using free dextranase. Furthermore, our results indicated that different sizes of nano-TiO_2_ could be chosen to adjust hydrolysate composition. Immobilized dextranase, therefore, has good application prospects in the food-processing industry and in the preparation of prebiotics.

The hydrolysates of immobilization dextranase have changed considerably. Isomaltotriose and isomaltopentraose were the main products (over 90% in total) of the free dextranase. However, isomaltotetraose was the main product (over 70%) of the immobilization one, and isomaltopentraose was not detected. A study found that immobilization of dextranase with alginate was able to synthesize isomalto-oligosaccharide, which is a prebiotic with significant effect [10]. Our results also showed that different particle sizes of TiO_2_ used to immobilize dextranase, might change the hydrolysates. As Figure 6c showed, the peak of isomaltopentaose can be found when the TiO_2_ size is 50 nm. The peak of isomaltotriose and isomaltotetraose can be seen if the TiO_2_ size is 5 nm and 30 nm, respectively. The results suggest that different sizes of nano carriers might change the composition of products. Different polymerization of isomalto-oligosaccharides has expressed diversity of function. They have been applied in the fields of nutrition, health care, and disease prevention and control [54]. The juices and candies containing added oligosaccharides could be consumed by patients of diabetes, as they have the advantage of promoting insulin secretion. Moreover, as a prebiotic, they can regulate the intestinal microenvironment, to improve the immune system [55].

### 3.6. Reproducibility and Storage of Immobilized Dextranase

We investigated the reusability and storage of immobilized dextranase. As shown in Figure 7, immobilized dextranase maintained >50% of its enzymatic activity at the seventh usage, indicating that it can be recycled repeatedly several times. Results from the thermostatic storage experiment revealed that the activity of immobilized dextranase was 58% after 7 days at 25 °C.

The immobilized dextranase could maintain over 70% activity at the 50th reusage. Miona G. Miljković et al. studied the immobilization of both TiO_2_/5-ASA/GA and TiO_2_/GOPTMS carriers for dextranase, and found that the immobilized dextranase on TiO_2_/5-ASA/GA retained about 70% after the fifth reuse cycle, whereas TiO_2_/GOPTMS retained only 15% of enzyme activity [8]. Our results showed the immobilized dextranase on TiO_2_ has good repeatability and can be used for future applications.

### 3.7. Adsorption Kinetic Study

It can be seen in Figure 8 that the PFO kinetic correlation coefficient (R^2^ = 0.81) for the adsorption of dextranase by TiO_2_ nanoparticles deviates significantly; therefore, it cannot be used to describe the complete adsorption process. The parameters of the PSO kinetic model showed R^2^ > 0.99 (R^2^ = 0.99) and it can be seen in Table 2 that the theoretical value of *q_e_* (3.94 mg/g) was also similar to the experimental value (3.99 mg/g). Therefore, the adsorption of dextranase by TiO_2_ nanoparticles was more in line with the PSO kinetic reaction.

The results showed adsorption amount Q_e,ecp_ was 3.99 mg/g, whereas Q_e,2_ obtained according to the PSO model theory was 3.94 mg/g. Through comparison, the experimental adsorption capacity is higher than the theoretical adsorption capacity, and the correlation coefficient R^2^ (0.99) obtained from the PSO model is closer to 1 than the PFO model. So, the PSO model is more suitable for the adsorption process of dextranase immobilization. Lim Wen Yao et al. also found the PSO model to be more suitable for the adsorption process, when they investigated the immobilization efficiency of lipase as a biocatalyst on graphene oxide [56].

## 4. Conclusions

We performed a feasibility study on the immobilization of dextranase with nano-TiO_2_. The optimum particle size of nano-TiO_2_ that was fixed for this study was 30 nm. The best immobilization efficiency was achieved at 35 mg of TiO_2_ nanoparticles. The immobilization efficiency increased with extended incubated time. The optimum immobilization temperature and pH were 25 °C and 7.0, respectively. The enzymatic properties of immobilized dextranase changed slightly. The optimum temperature of immobilized dextranase decreased to 30 °C from 40 °C of free dextranase. The stability of immobilized dextranase improved considerably at lower pH, and it showed higher activity under acidic conditions. The results from FT-IR, XRD, and FEG-SEM, demonstrated that dextranase was immobilized on the surface of TiO_2_ nanoparticles. In the hydrolysates, a higher degree of polymerized isomaltotetraose (78.69%) was obtained after 30 min of incubation. The immobilized dextranase retained >50% of its hydrolytic capacity at the seventh usage. Moreover, its activity was 58% after 7 days at 25 °C, indicating the recyclability of the immobilized dextranase. The adsorption of dextranase by TiO_2_ nanoparticles conformed to the PSO reaction process. The immobilized dextranase is conducive to use owing to its higher durability, thereby increasing its practical applications in industry, and highlighting its application prospects.

## Figures and Tables

**Figure 1 nanomaterials-13-01065-f001:**
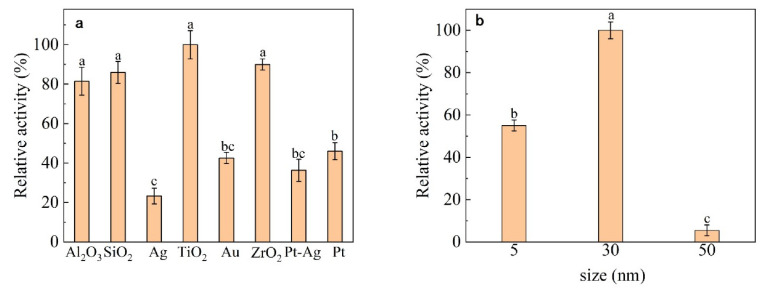
Nanomaterial selection: (**a**) different nanomaterials; (**b**) different particle sizes of TiO_2_. On the bar chart, the different lowercase letters show that there are significant difference (*p* > 0.05).

**Figure 2 nanomaterials-13-01065-f002:**
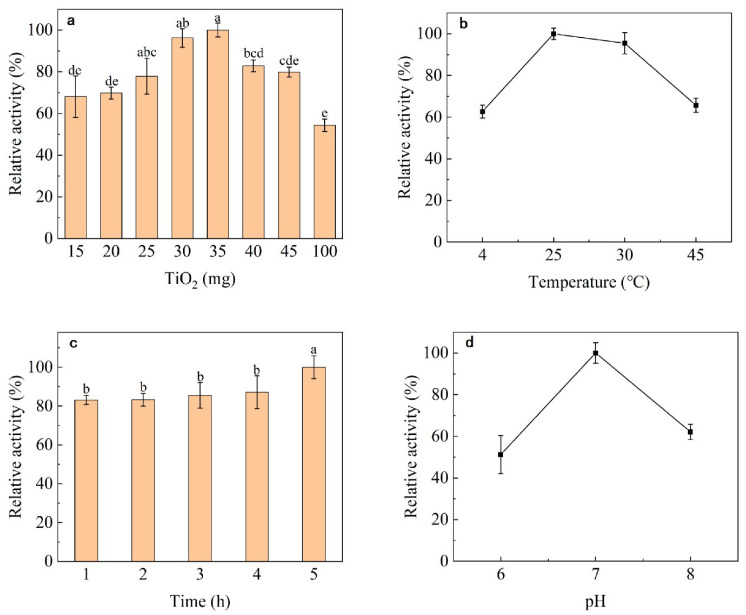
Effect of different conditions on enzyme immobilization: (**a**) TiO_2_ content, (**b**) temperature, (**c**) time and (**d**) pH. On the bar chart, the different lowercase letters show that there are significant difference (*p* > 0.05).

**Figure 3 nanomaterials-13-01065-f003:**
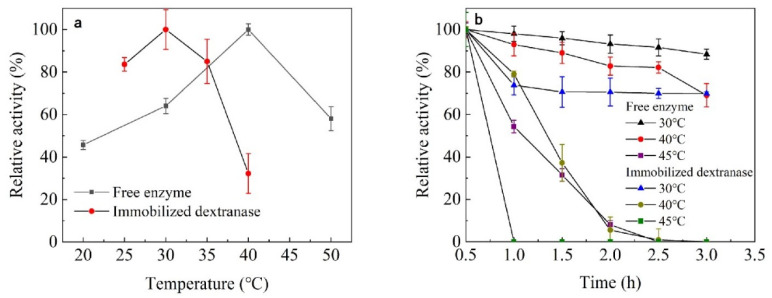
(**a**) Optimum temperature, and (**b**) temperature stability.

**Figure 4 nanomaterials-13-01065-f004:**
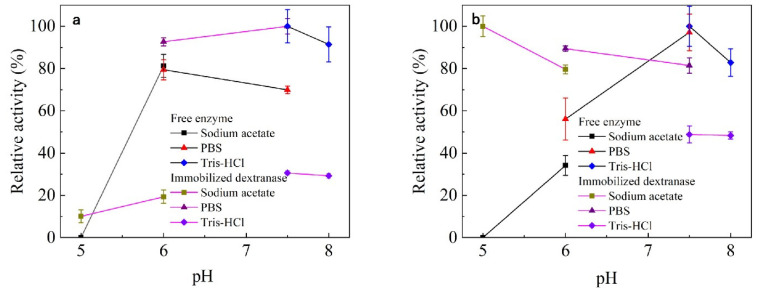
(**a**) Optimal pH, and (**b**) pH stability.

**Figure 5 nanomaterials-13-01065-f005:**
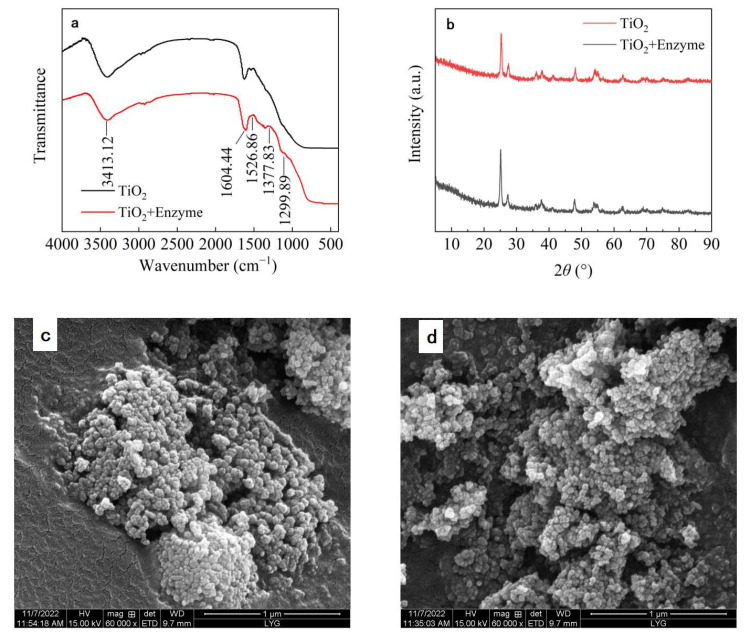
(**a**) FT-IR spectrum of the immobilized enzyme, (**b**) XRD patterns of the immobilized enzyme, (**c**) FEG-SEM images of TiO_2_, (**d**) FEG-SEM image of immobilized dextranase.

**Figure 6 nanomaterials-13-01065-f006:**
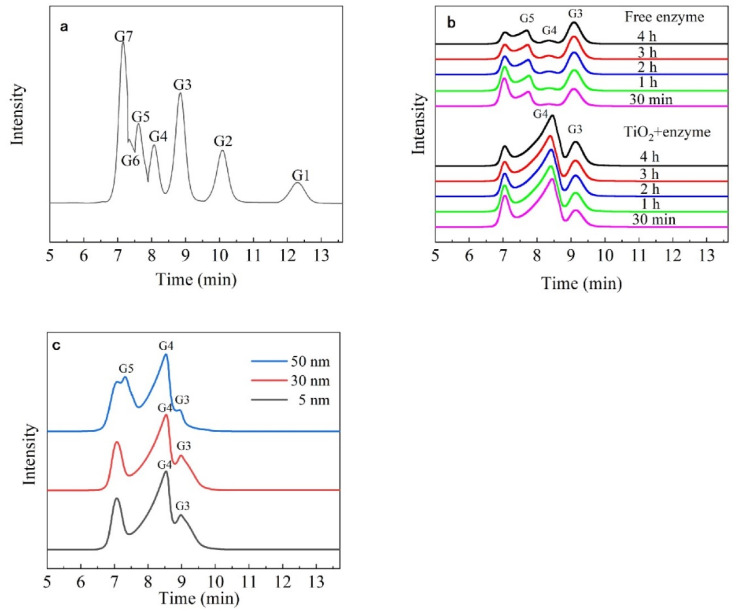
(**a**) Standard oligosaccharides G1–G7: glucose, isomaltose, isomaltotriose, isomaltotetraose, isomaltopentaose, isomaltohexose, and isomaltoheptaose (**b**) free enzyme and immobilized dextranase enzymatic products at different reaction times, (**c**) analysis of the 1 h digestion products of 5 nm, 30 nm, and 50 nm TiO_2_-immobilized dextranase.

**Figure 7 nanomaterials-13-01065-f007:**
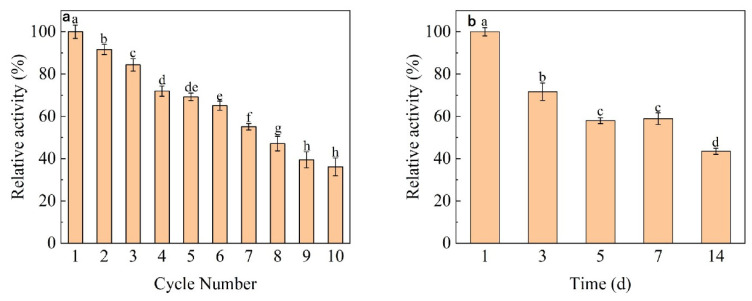
(**a**) Immobilized dextranase reusability assay. (**b**) Immobilized dextranase storage assay. On the bar chart, the different lowercase letters show that there are significant difference (*p* > 0.05).

**Figure 8 nanomaterials-13-01065-f008:**
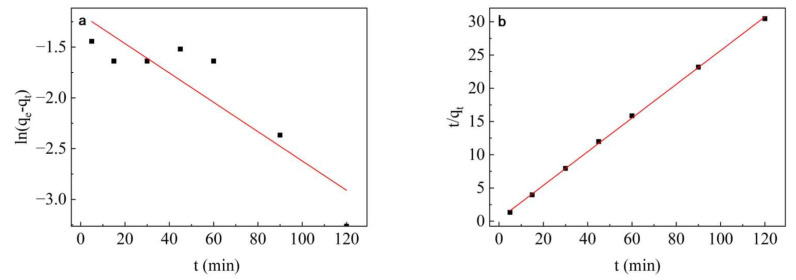
(**a**) Pseudo–primary (PFO) kinetic model, and (**b**) pseudo–secondary (PSO) kinetic model.

**Table 1 nanomaterials-13-01065-t001:** Analysis of the ratio of the hydrolysates of dextranase.

Hydrolysis Time (h)	Hydrolysates (%)		
	Isomaltotriose	Isomaltotetraose	Isomaltopentaose
Pure enzyme			
0.5	58.12 ± 1.94	6.03 ± 0.75	35.85 ± 5.71
1	61.95 ± 3.31	4.65 ± 1.60	33.40 ± 5.54
2	62.98 ± 4.96	5.4 ± 1.57	31.62 ± 4.31
3	62.34 ± 5.76	5.75 ± 1.55	31.91 ± 3.41
4	62.06 ± 5.45	5.93 ± 1.66	32.01 ± 3.42
Immobilized enzymes			
0.5	21.31 ± 2.05	78.69 ± 4.95	-
1	23.23 ± 1.58	76.77 ± 4.59	-
2	25.91 ± 3.07	74.09 ± 4.13	-
3	26.66 ± 3.43	73.34 ± 3.66	-
4	26.88 ± 3.25	73.12 ± 3.39	-

**Table 2 nanomaterials-13-01065-t002:** Kinetic model parameters.

Q_e,ecp_ (mg/g)	Pseudo First-Order Dynamics			Pseudo-Second Order Dynamics		
	K_1_ (min^−1^)	Q_e,1_ (mg/g)	R^2^	K_2_ (g·mg^−1^·min^−1^)	Q_e,2_ (mg/g)	R^2^
3.99	0.03	0.308	0.81	0.21	3.94	0.99

## Data Availability

The datasets generated during and/or analyzed during the current study are available from the corresponding authors upon reasonable request.

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
