# Peer review of "Immobilization of Dextranase Obtained from the Marine Cellulosimicrobium sp. Y1 on Nanoparticles: Nano-TiO2 Improving Hydrolysate Properties and Enhancing Reuse"

_nanomaterials, 2023, doi:10.3390/nano13061065_

Round 1

Reviewer 1 Report

The authors try to immobilized efficiency of dextranase on nano-TiO2 under different conditions and study the changes in the enzymatic properties
and hydrolysates of immobilized materials and explored the reproducibility and storability of the immobilized dextranase.

In order to be published the authors should improve the material with the following:

1. The introduction has to be improved with relevant data in order to explain the results.

2. when you describe what you will study, please be more specific.

3. please try to complete the data regarding the synthesis there are some data missing.

4. please revise: "The immobilized materials showed a decrease in the pores between TiO2 nanoparticles and an increase in agglomeration (Figure 5c, d), indicating the successful immobilization of dextranase on TiO2 nanoparticles. "

how can you tell from this pictures the dimensions of pores ? eventually from literature we can observe that pores are decreasing but add this with citation.

5. usualy when something organic is added the amorphicity is higher, how can you explain ? please add the XRD without vertical translation.

6. i suggest the authors to change the title in order to conclude what you are trying to do in this article.

7. some parts of english should be improved.

Reviewer 2 Report

The manuscript subject of the present review deals with an inetersting and scientifically significant area of nanotechnology, namely the efficiency and applicability of enzyme-immobilized nanoparticles. However, the paper needs major revision due to a number of inaccuracies, errors, and weak points:

1. All used chemicals and reagents have to be included and described in section Materials, including Al2O3, SiO2, Ag, Au, ZrO2, Pt-Ag, and Pt, Tris-HCl buffer!

2. UV/Vis spectrophotometry has to be added as an applied method together with the type and manufacturer of the spectrophotometer.

3. What does "sterile water"mean?

4. The information in Subsections 2.2.3.1 and 2.2.3.2 actually coincides with that in 2.2.2.3.

5. How were the immobilized nanoparticles dryed?

6. Lines 110-112 probably have to be at the beginning of section 2.2.4?

7. Lines 114-115: What does 1-7 oligosaccharides mean?

8. How was sonication performed - parameters, apparatus?

9. The description of the HPLC analyses has to be improved: mobile pahse content, conditions, detection time, apparatus!

10. Section 2.2.5 has to be fully revised!

11. Line 135 - Lagergren!

12. Lines 141, 142 the units of the the capacity and rate constants have to be added.

13. Statistical analyses and signigicance of the experimental results is missing!!!

14. The results and discussion section has to be revised - more discussion is necessary and comparison with similar scientific studies already published.

15. The conclusion made within lines 200-203 does not sound scientifically proven as it is difficult to withdraw it on the bases of the SEM images presented.

16. Section 3.7: What does the better applicability of the PSO kinetics model is associated with and why the authors modelled the adsorption kinetics of the enzyme?

17. The Conclusions section has to be revised. The conclusions have to follow the order of the investigations.

18. The sections dealing with the selection of the nanomaterial have etiher to be improved with more data concerning the specific inorgnic materials: physicochemical properties, pros and cons, etc., or to be excluded from the manuscript. Besides, the fact that one of the goals of the study is selection of an appropriate nanomaterial has to be included in the title of the paper. 

Reviewer 3 Report

I do not consider the article submitted by the author to be suitable for publication in the journal "Nanomaterials". Overall, there are no fundamental problems with the method and research, but the novelty of the research is highly questionable and there is no justification for choosing the selected nanoscale materials (except for TiO2). Furthermore, it is not clear what justified the change in the test conditions (pH, temperature, condensation) that formed the space of the parameter. The scope and scientific content of the research work carried out is much closer to a well-done student work, which was prepared on already known foundations, and the content of its novelty is not significant. Given all this, I don't see the article as suitable for displaying in a Q1 level journal.

Reviewer 4 Report

The manuscript presents an interesting topic and is well written.

The studies are conducted rigorously and the data is clearly presented,

Nonetheless I think the manuscript would benefit if the following minor adjustments will be made:

Firstly, the authors should improve the introduction section and take into consideration similar studies performed by other research groups emphasizing the novelty of the present study. Moreover, the authors should also include discussions regarding their findings compared to other reported studies.

Secondly, the authors should add to the FTIR spectrum the second derivative. This will allow a better understanding of the vibrational bands form the spectrum.

Thirdly, the authors should improve the resolution of some of the figures (5,7 and 8)

Round 2

Reviewer 3 Report

The authors have made efforts to correct the article based on the corrections requested by other reviewers. Now I find the article suitable for publication in the journal of Nanomaterials.